# Deep Trench Isolation and Inverted Pyramid Array Structures Used to Enhance Optical Efficiency of Photodiode in CMOS Image Sensor via Simulations

**DOI:** 10.3390/s20113062

**Published:** 2020-05-28

**Authors:** Chang-Fu Han, Jiun-Ming Chiou, Jen-Fin Lin

**Affiliations:** 1Department of Mechanical Engineering, National Cheng Kung University, Tainan 701, Taiwan; cfhan@mail.ncku.edu.tw (C.-F.H.); tom0090117@gmail.com (J.-M.C.); 2Center for Micro/Nano Science and Technology, National Cheng Kung University, Tainan 701, Taiwan

**Keywords:** deep trench isolation, inverted pyramid array, optical efficiency, near-infrared

## Abstract

The photodiode in the backside-illuminated CMOS sensor is modeled to analyze the optical performances in a range of wavelengths (300–1100 nm). The effects of changing in the deep trench isolation depth (DTI) and pitch size (d) of the inverted pyramid array (IPA) on the peak value (*OE_max_*_._) of optical efficiency (OE) and its wavelength region are identified first. Then, the growth ratio (GR) is defined for the *OE* change in these wavelength ranges to highlight the effectiveness of various DTI and d combinations on the *OEs* and evaluate the *OE* difference between the pixel arrays with and without the DTI + IPA structures. Increasing DTI can bring in monotonous *OE_max_*_._ increases in the entire wavelength region. For a fixed DTI, the maximum *OE_max_*_._ is formed as the flat plane (*d* = 0 nm) is chosen for the top surface of Si photodiode in the RGB pixels operating at the visible light wavelengths; whereas different nonzero value is needed to obtain the maximum *OE_max_*_._ for the RGB pixels operating in the near-infrared (NIR) region. The optimum choice in d for each color pixel and DTI depth can elevate the maximum GR value in the NIR region up to 82.2%.

## 1. Introduction

Demand for solid-state image sensors has grown due to increasing requirements for mobile imaging, digital still and video cameras, surveillance, monitoring, and biometrics. Complementary metal-oxide semiconductor (CMOS) image sensors have been adopted in numerous products, such as computer cameras, optical mice, and mobile phones. An important advantage of CMOS image sensors over charge-coupled device (CCD) sensors is the ability to integrate sensing with analog and digital processings down to the pixel level. The technology, design, and performance limits of CMOS image sensors, along with recent developments and future directions, have been reviewed [1]. A comparison of passive and active pixel schemes for CMOS visible images was made in the study of Kozlowski et al. [2]. An active pixel sensor (APS) integrated in the standard CMOS process technology was found to be superior to a passive pixel sensor.

CMOS image sensors can have pixel sizes of 2 μm × 2 μm or even lower (below 1 μm) [3]. A major challenge is the suppression of all parasitic charge exchange between neighboring pixels (crosstalk), which can lead to loss in image quality. Several pixel isolation architectures have been proposed for suppressing crosstalk, including the deep trench structure [4], in which the Si-SiO_2_ interface is used as a barrier against electron diffusion. In the study of Tournier et al. [5], deep trench technology for CMOS image sensors was developed for 1.4-μm pixel front-side illumination technology. A comparison with other pixel isolation architectures showed that deep trench isolation (DTI) is a good approach for suppressing electrical crosstalk. A 1.12-μm backside-illuminated CMOS sensor with backside DTI was demonstrated in the study of Kitamura et al. [6]. A 50% reduction in crosstalk was achieved.

The Bayer array is a commonly used color filter array [7]. The color green is adopted as the representative of luminance because the luminance response curve of the human eye peaks at around the frequency of green light (550 nm). Various light-trapping structures based on crystalline Si materials, including nanowires, nano rods and holes, and inverted pyramids, have been investigated through numerical simulation [8,9,10]. The nanowire arrays with moderate filling ratio have much lower reflectance compared to thin films [8], and nanohole arrays have great potential for efficient solar photovoltaics [9]. The effects of inverted pyramid nanostructures correspond to a reduction in silicon mass by two orders of magnitude, pointing to the promising future of thin crystalline silicon solar cells [10]. Experimental results show that crystalline silicon (c-Si) thin films with a thickness of less than 10 μm can absorb light as well as 300-μm-thick flat c-Si substrates when they are decorated with inverted nanopyramid structures [11]. Other advantages of using inverted nanopyramids are that the interface area in the silicon photodiode is 1.7 times larger than that of a flat surface, which can minimize surface recombination losses, and that the fabrication of the structures is easy and cost-effective. The study of Yokogawa et al. [12] reported that the infrared sensitivity of a back-illuminated CMOS image sensor is enhanced at certain wavelengths by a two-dimensional diffractive inverted pyramid array (IPA) on crystalline silicon and DTI. NIR images taken by a camera equipped with a C-mount lens showed a 75% sensitivity enhancement in a wavelength range of 700–1200 nm with negligible spatial resolution degradation.

The study of Vaillant et al. [13] developed a methodology for the optical simulation of CMOS image sensors. With the use of an incoherent summation of plane wave sources and Bloch periodic boundary conditions, simulation results were obtained with Lumerical FDTD Solutions software. The results of rigorous electromagnetic broadband simulations applied to CMOS image sensors as well as experimental measurements have been presented [14], with reported improvements in the spectral responses of antireflection and an optical stack reduction. An electromagnetic simulation tool based on FDTD Solutions [15,16], available from Lumerical [17], was used to describe light propagation and photon collection inside a pixel. A finite-difference time-domain (FDTD)-based optical simulation mode for describing the optical performance of CMOS image sensors considering diffraction effects has been proposed [18]. Periodic conditions are used to limit the size of the simulated model and thus reduce both memory usage and computational time.

The DTI depth (DTI) and the pitch size (d) in the IPA structure can affect the crosstalk and photon collection inside the RGB pixels. The combined effect of these two controlling factors on optical efficiency (OE) in a wide range of wavelengths including the NIR region has seldom been reported [12]. The reported experimental and simulational results for visible light wavelengths are quite limited and insufficient to determine the optimal conditions of these two factors for the peak optical efficiency (*OE_max_*). Numerical analyses made for the single-factor effect of DTI and IPA structures show that *OE_max_* values in a wide wavelength ranges are elevated, and especially significant for those created in the NIR wavelengths. The finding of the optimum combination of DTI and IPA structures to obtain the maximum values of *OE_max_* (Max *OE_max_*) in these visible-light and NIR regions through the efficient and accurate way of numerical analyses becomes the purpose and novelty of this study. In this study, two green pixels in an array are positioned in a diagonal form to separate the red and blue pixels on their two sides. A three-dimensional (3D) solution for a Bayer filter array with the red/green/blue (RGB) pixels in the backside-illuminated CMOS image sensor was thus obtained using the FDTD Solutions software to analyze the variations of its quantum efficiency (QE) with the light wavelength in a range of 300~1100 nm. The *QE* results are compared to previously reported experimental values to verify the trustworthiness of the software modules and boundary conditions, and the material optical properties applied to the present study. Then, the modules are extended to predict the *OE_max_*_._ value of the RGB pixels for a wide range of DTI and d value, and at the wavelengths between 300 to 1100 nm. The 3D profile for the *OE* ratio, which is defined on the basis of the *OE* values created in the pixel arrays without and with the DTI + IPA structures, is provided favorable for the choices of DTI and the d of IPA with the Max. *OE_max_*_._ values formed in the four wavelength regions (450~460 nm for blue pixel; 530~540 nm for green pixel; 610~620 nm for red pixel; and 810~860 nm for NIR).

## 2. Simulation Methodology

### 2.1. FDTD Application

The FDTD software [17] is usually applied to solve the time-dependent solutions for the Maxwell-Boltzmann differential equations. A change in the electric field (E) is linked with a change in the magnetic field (H) across the space. Generally, FDTD engines use the Yee algorithm to solve these equations [15,19]. The periodic boundary condition considers the simulated structure to be infinitely repeated, like a periodic crystal, which allows us to reduce the simulated domain to its elementary zone. The present work adapts the FDTD Solutions software to develop a relevant pixel model while keeping the computational time and memory usage to be reasonable. The optical software Lumerical FDTD Solutions [17] was adopted because of its high efficiency as the calculation engine.

Figure 1a shows the schematic diagram of a backside-illuminated CMOS image sensor which is generally composed of the three parts: RGB pixels, metal wiring layer and Si substrate. The pixel transmission (*T*) part records the normal component of the Poynting vector, from the top surface (into Si photodiode) to the bottom surface (through the RGB pixels) of the Si photodiode (see Figure 1b). The Poynting vector (${\vec{P}}_{}$ unit: W/m^−2^) is obtained as the cross product of the electric and magnetic fields:(1)$$\vec{P}=\frac{1}{2}\left[\vec{E}\times \vec{H}\right]$$

The simulator propagates the electric field in the primary grid and the magnetic field in the secondary grid. The steady simulational solution is used to interpolate the components of the magnetic field on the primary grid, making it possible to calculate the Poynting components everywhere inside the simulational region. If we want to quantify the overall optical performance, it is necessary to calculate the absorbed energy in the silicon photodiode. To calculate the power absorbed in each pixel, we integrate Poynting vector over the depletion region of the pixel. The optical efficiency (OE) is defined as the fraction of the power incident onto the pixel, that is absorbed in the depletion region of the pixel [17,18]:(2)$$\mathrm{Optical}\;\mathrm{efficiency}\;(\mathrm{OE})=\frac{\mathrm{Absorbed}\;\mathrm{power}}{\mathrm{Source}\;\mathrm{power}}=\frac{{\left(T\right)}_{\mathrm{into}\;\mathrm{Si}}-(T{)}_{\mathrm{through}\;\mathrm{pixel}}}{{\left(T\right)}_{\mathrm{into}\;\mathrm{Si}}}$$

The OE of a material can be obtained from the FDTD simulations [17].

### 2.2. Photodiode Analysis Developed for Optical Efficiency Evaluations

A CMOS sensor pixel can generally perform photon collection, photon-electron conversion, and reading. Every pixel thus comprises several complex components. The components above the silicon photodiode have different optical effects in a backside-illuminated CMOS sensor pixel array, they are the layers including the Si_3_N_4_ antireflection coating which improves the efficiency since by a less light lose due to its small reflection, the color filter (CF) pixel array (Bayer pattern) [7] which consists of colored resin for decomposing the white light into its red, green, and blue primary colors, and the microlenses (SiO_2_) above pixels which are applied to focus the incident light onto the photodiode. To evaluate the illumination of the CMOS detector, plane waves, rather than focused beams, are generally applied as the light source. In the present study, we focus on the effects of combing different DTI and d of the IPA structure on the OEs arising in the silicon photodiode layer of the RGB pixels only. The effect of the reflection of the transmission by the metal wiring layer below the photodiode will be discussed in the later section. Therefore, the FDTD simulations are carried out for the photodiode domains (excluding the microlenses and antireflection coating) whose top surface is exposed to the plane waves.

#### 2.2.1. Photodiode in RGB Pixel Array

In the present study, a RGB pixel array in a backside-illuminated CMOS image sensor is adopted for the OE evaluations using the numerical scheme in Lumerical FDTD Solutions software for photon simulations. The simulations were carried out for the 3D domain of one RGB pixel array only. Figure 1b shows the schematic diagram of the lateral surface of a RGB pixel array used for the OE analysis. In this study, the total-field scattered-field plane wave with an incident angle of 0° is assumed to have the point sources uniformly distributing over the RGB pixel array. Therefore, only one Bayer filter array with the RGB pixels is adopted as the domain of simulations. The top surface of the array has an area of 2000 nm (length) × 2000 nm (width), and about 1000 nm (length) × 1000 nm (width) for every pixel. The DTI structures have been prepared in the areas with either a cross (“+”) form pattern to separate or a square form (“□”) to frame the four pixels, as shown in Figure 1b. The widths for the “+” and “□” DTI structures were set to 100 and 50 nm, respectively. The thicknesses for the components of microlenses, color filters, antireflection coating, and photodiode Si substrate are 360, 500, 50, and 6000 nm, respectively. The microlenses had a conic form with a radius of curvature of 1 μm and a conic constant of-1.

In the present study, an IPA with various dimensions was inserted between the antireflection coating and silicon photodiode in order to improve the OE of the photodiode. If the pitch size of the IPA structure on Si photodiode is much smaller than the wavelength of interest, the effective refractive index between Si photodiode and the upper layer has an intermediate value, which is well-known as an anti-reflection coating or refractive index gradient structure [20]. Meanwhile, a nonzero value is expected to have the diffraction and anti-reflection effects, thus elongating the optical path length within the Si layer [12]. An increment of d generally results in an absorption decrease and reflection increase slightly. In the simulations, Si (100) is adopted as the crystallographic direction of the surface before etching. The angle between the horizontal and etched surfaces after wet etching was obtained theoretically as 54.7° (see Figure 2). *d* is one of the controlling factors for the pyramid height ($\bar{z}$) and the OE of the pixel array.

#### 2.2.2. Material Settings

To evaluate the OE, the data of the complex refractive index, *N_complex_*, for the components of the RGB pixel array at various wavelengths should be given as the inputs of the optical properties. The complex refractive index is expressed as:(3)$$N_{complex}=n+ik$$
where *n* is the index of refraction, and *k* is the extinction coefficient. The *n* and *k* values evaluated for a wavelength range of 300–1100 nm for Si, SiO_2_, and Si_3_N_4_ are provided in the Lumerical FDTD Solutions software [17]. The *n* and *k* data evaluated for the blue, green, and red filters at the wavelengths, 300–1100 nm, are also provided [21,22]. The optical properties (*n* and *k*) for the materials of the microlenses, color filter, and antireflection coating in the FDTD software for CMOS sensors are shown in Table 1.

#### 2.2.3. Incident Light Source Settings

A total- and scattered-field light sources with the plane waves are applied to carry out the OE evaluations. Light source is generally produced by a halogen lamp and a monochromator. The light source was partly guided to a calibrated photodiode to control its intensity. A f-number diffuse light incident on the sensor can be created via the light passing through a diffuser and a f/D controller. In this study, the diffuser is assumed to be a sufficiently large size such that every pixel in the sensor is expressed to a uniform illumination with the same *f* value in the simulations. Then, only the “Lumerical” module in the FDTD software meets this requirement. In this study, the light source plane was set at a 10-μm distance above the top surface of the pixel array. They were chosen to be “Periodic” in the source setting, with an incident angle of 0° (see Figure 1).

In the present study, the wavelengths of 300–1100 nm were set as the range for the OE simulations. The simulation scheme was applied to obtain the OE solutions and investigate the effects of the controlling factors (DTI and d) on OE.

#### 2.2.4. Boundary Condition Settings

In the FDTD analyses, a uniaxial anisotropic perfectly matched layer was assumed for the boundaries normal to the *z* direction (see Figure 1b). This assumption was made to avoid the multiple reflections from the pixel components above the Si photodiode. Regarding the light passing through the Si photodiode, the normalized transmission ($\bar{T}$) in the position of “through the pixels” is evaluated to have the quantify about 0.5~0.8% of the quantify of “into the Si photodiode” in the FDTD simulational results. Thus, the optical reflection by the metal wiring layer below the Si photodiode can be neglected. The reflection by the electrical components below the Si photodiode is thus excluded from the present analyses. As for the boundaries in the x-y plane, periodic boundary conditions were set for the four edges of each pixel in the Bayer filter array.

#### 2.2.5. Quantum Efficiency for CMOS Sensors

*QE* is defined as the ratio of the collected electrons to the incoming photons, it is affected by the illumination conditions, the objective lens, the *OE* of the optical stack of the image sensor, and the collection efficiency of electronics. A full calculation of the *QE* involves both optical and electrical modeling. It is generally classified as the external quantum efficiency (*EQE*) and the internal quantum efficiency *(IQE*). The *IQE* is the ratio of collected charge number to the number of incident photons, and is dimensionless. In the electrical stimulation of Lumerical software [17], the *IQE* can be calculated by a Green’s function approach for an arbitrary optical generation rate by determining a spatially-varying weighting function that represents the collection probability for a photo-generated electron-hole pair. Assuming that each absorbed photon generates an electron hole pair, *EQE* is simply the product solution of *OE* and *IQE* and it is written as [17]:(4)QE=EQE=IQE × OE

The *OE* solutions can be obtained from the FDTD simulations if the controlling parameters including the dimensions of pixel array, DTI, *d*, and *IQE* are available.

## 3. Results and Discussion

The *QE* data reported for the wavelengths in a range of 300 to 1100 nm were obtained from the experimental measurements [21], and they are shown in Figure 2a. The solid and dashed curves are provided for the experimental data [21] and the FDTD solutions for the 1.0 μm pixel array, respectively. These three sets of solid and dashed curves are presented for the blue, green and red CF pixels operating in a wavelength range of 300 to 1100 nm. By the data of *IQE* = 0.87 and DTI = 2000 nm, the *QE* results from the measurements and the FDTD analyses are found very close each other and valid for the entire range of wavelengths. This characteristic implies that the FDTD simulations and the set boundary conditions can provide accurate predictions of the *OE* results for the RGB pixels if the *n* and *k* values of all pixels operating at various wavelengths are available. This software module plus the optical parameters (*n* and *k*) are extended to the CF pixels operating at the wavelengths between 300 and 1100 nm in order to investigate the effects of pixel width, DTI and d in the IPA structure on *OE*. The *OE* solutions for the three CF pixels are shown in Figure 2b. The variations of *OE* for the two structures with 1.0 μm and 0.9 μm pixels with DTI = 6000 nm, have shown quite similar patterns, and the *OE_max_*_._ values for the RGB pixels show a small difference between these two pixels and they are risen by increasing the pixel width. Figure 2c shows the *OE* profiles for the RGB pixels in the 0.9-μm-pixel arrays with the solid curves defined for the pixel array without DTI and the dashed curves for the pixel array with DTI = 6000 nm. The nonzero DTI depth can elevate the *OE_max_*_._ values for the blue, green and red pixels arising in the visible light wavelength region (*OE_max_*_._: 81.33 → 85.32% for blue, 82.90 → 87.23% for green, and 73.21 → 76.17% for red), and the *OE_max_*_._ values of these three CF pixels operating in the NIR wavelength region (>780 nm). The *OE* results of the RGB pixels for the 0.9-μm-pixel arrays with the DTI and DIT + IPA structures are shown in Figure 2d, the DTI and d are 6000 nm and 300 nm, respectively. They reveal the characteristic that the OE_max._ values of the blue, green and red pixels in the visible light wavelength (*λ*) region (300 nm ≤ *λ* ≤ 780 nm) are slightly lowered by adding the IPA structure to the specimen with DTI = 6000 nm; however, the *OE_max_*_._ values formed at the wavelengths between 810 and 860 nm (in the NIR region) are significantly elevated by adding an IPA structure with *d* = 300 nm. It is thus concluded that the use of an IPA structure can raise the *OE_max_*_._ values in the NIR region significantly, especially the one arising at the red pixel.

The *OE_max._* values for the RGB 0.9-μm pixels created in the visible light wavelength region are evaluated for various DTI value shown in Figure 3a,d values, shown in Figure 3b. In either of these two controlling factors used as the variable, (*OE_max._)_green_* > (*OE_max._)_blue_* > (*OE_max._)_red_* is obtained. Monotonous increases in *OE_max._* are presented in every CF pixel as increasing the DTI although their growth rate is small. This behavior reveals that an increase in DTI can reduce the crosstalks of these three CF pixels arising at the visible light wavelengths. It should be mentioned that the wavelength associated with the *OE*_max._ is dependent on the DTI, and thus the wavelengths of *OE_max_*_._ for every CF pixel are varied in a narrow range, rather than a fixed value. The results shown in Figure 3b are presented to evaluate the effect of pitch size (d) on the *OE_max._* values for the three CF pixels operating in the visible light wavelengths. *d* = 0 nm is defined for the top surface of silicon photodiode to be a flat plane (without IPA structure) perpendicular to the plane waves. In order to explain the OE*_max._* behavior demonstrated in Figure 3b, the reflection created in the Si photodiode layer can be evaluated as a function of d. Figure 3c shows the reflection results in these three wavelength regions. Reflections are monotonically elevated by increasing the pitch size although their value is quite small (<1.5%). An increment of reflection (%) will bring in the transmission and absorption decreases, thus resulting in the OE*_max._* decline.

The effect of DTI depth on the electric field distribution can be evaluated for the 0.9-μm green and red pixels. The silicon substrate thickness was set to be 6 μm and the DTI width was 100 nm in the simulations. The DTI depth was designed to have 0, 2000, 4000, and 6000 nm, respectively. To investigate the *OE* behavior demonstrated at the NIR wavelengths, plane light waves with $0^{o}$ as the incident angle are applied to predict the cross-sectional distribution of the electric field intensity (EI) which is usually regarded as the performance of optical absorption in the pixel. In this study, 850 nm is chosen as the simulational wavelength because the wavelengths corresponding to the *OE_max_*_._ values created in the RGB pixels are quite close to this wavelength. The Lumerical results for the DTIs, 0 and 6000 nm, are shown in Figure 4a,b, respectively. The origin of the simulational domain was located at the bottom surface of the Si photodiode as well as the left corner of the red pixel. Because 850 nm is now the wavelength in the NIR, photons absorbed by the red CF pixel are thus much more than those by the green pixel. As the EI results show in Figure 4a,b, a nonzero DTI used can elevate the maximum EI value (EI_max._) and extend the comparatively higher EI distribution to a wider area of the red pixel; Conversely, it brings a reduction of EI_max._ in the green pixels. Obviously, an increase in the DTI depth can reduce the crosstalks between the two adjacent pixels efficiently. The above effects are generated by filling the SiO_2_ in the DTI structures, which will occur the total internal reflection at the interface to the silicon and lead to more light trappings.

An IPA structure with d = 300 nm was introduced to the top surface of the silicon photodiode to increase the exposed area for light incidences. DTI = 0 and 6000 nm were used for the 0.9-μm pixels. The aspect ratio of the IPA structure for d = 300 nm was fixed at 0.708 (=1/Tan(54.7°)) because of the facet angle formed between c-Si (111) and (100) [12]. The numerical EI results for DTI = 0 and 6000 nm are shown in Figure 5a,b, respectively. They are obtained for the wavelength of 850 nm. The results in Figure 4 and Figure 5 are provided to investigate the effect of IPA structure on the EI distributions in the red and green pixels. The combined effect of nonzero DTI and d can elevate the EI_max._ of the red pixel but reduce the EI_max._ of the green pixel of the specimens simply with the DTI structure only.

The *OE* results shown in Figure 3 reveal that the geometries including pixel width, d and DTI become the controlling factors for the *OE* evaluations at various wavelengths. In the present study, the combined effect of various DTI depths and pitch size for the *OE_max_*_._ values of the blue, green, and red pixels are shown in Figure 6a–c, respectively. 0 ≤ *d* ≤ 1000 nm and 0 ≤ DTI ≤ 6000nm are set to obtain the *OE_max_*_._ solutions for the 0.9 μm RGB pixels. Each of these three *OE* curved surfaces show that either an increase in DTI or a decrease in d of the RGB pixels can lead to a rise of *OE_max_*_._ Therefore, the highest *OE_max._* denoted as Max. *OE_max_*_._ is usually present at the three CF pixels with DTI = 6000 nm and *d* = 0 nm. (Max. *OE*_max._)_blue_ = 85.3%, (Max. *OE_max_*_._)_green_ = 87.2%, (Max. *OE_max_*_._)_red_ = 76.2% are obtained for the visible light wavelengths in the regions of 450~460 nm (blue), 530~540 nm (green) and 610~620 nm (red), respectively. The *OE* profile for these three CF pixels also exist their *OE_max_*_._ in a wavelength region of 810~860 nm (in the NIR region). Figure 7d shows the *OE_max._* values of the blue, green, and red pixels respectively evaluated for this NIR wavelength region. These *OE_max_*_._ results indicate that the Max. *OE_max_*_._ values are arising at the conditions of DTI = 6000 nm in combination with d ≅ 400, 450 and 500 nm for the blue, green, and red pixels, respectively. (Max. *OE_max_*_._ = 42.2%)_blue_ and (Max. *OE_max_*_._ = 44.7%)_green_ and (Max. *OE_max._* = 50.1%)_red_ are obtained. Conclusions can be drawn as follows: (1) increasing DTI can bring in monotonous *OE_max._* increases in both the visible light and NIR wavelength regions, irrespective of the pitch size; (2) the uses of *d* = 0 nm and DTI = 6000 nm are advantageous for the RGB pixels operating in the visible light wavelength region with their Max. *OE_max_*_._; however, appropriate choices of nonzero d for the RGB pixels are needed to obtain their Max. *OE_max_*_._ arising at the wavelengths (810~860 nm) in the NIR region.

The *OE_max_*_._ data shown in Figure 6 are provided to evaluate the growth ratio (GR) of optical efficiency (*OE*) due to the use of different d and DTI. Define *SOE^DTI + IPA^* as the sum of the OE values arising in a specified wavelength range for the pixel array with various combinations of nonzero DTI and d. For example, the *SOE^DTI + IPA^* value in the 450~460 nm wavelength region is obtained to be the sum of (${\mathrm{OE}}_{\max.}^{\mathrm{DTI}+\mathrm{IPA}}$)_blue_ and the *OE^DTI + IPA^* values of the green and red pixels evaluated at the same wavelength. *S**OE^No DTI + IPA^* is also defined as the sum of the *OE* values similar to these defined for *SOE^DTI + IPA^* unless it is obtained for the array without the DTI and IPA structures. These two *SOE* parameters are defined valid for the wavelengths in the 450–460 nm, 530–540 nm, and 610–620 nm regions only. As to the *SOE^DTI + IPA^* and *SOE^No DTI + IPA^* parameters defined for the wavelength range between 810 and 860 nm, they are obtained to be the sum of the three *OE_max_*_._ values of the RGB pixels existing in this wavelength region. The growth ratio, *GR*, is thus expressed as:(5)$$GR=\frac{SOE^{DTI+IPA}-SOE^{No\;DTI+IPA}}{SOE^{No\;DTI+IPA}}$$

The *GR* value obtained from these two *SOE* values shows its definition basis characteristically different from those the *OE_max_* results shown in Figure 6. Figure 7a–c show the variations of *GR* with various d and DTI combinations for the blue, green, and red pixels, respectively; and Figure 7d shows the variations of *GR* in the RGB pixels working in the wavelength region between 810 and 860 nm. 4.9%, 4.2% and 6.6% are obtained as the maximum *GR* values for the blue, green, and red pixels respectively working in the visible light wavelength regions; and 82.2% is obtained to be the maximum *GR* value for the RGB pixels operating in the NIR wavelength region. In the blue and green pixels, there exist negative *GR* values in the conditions of comparatively larger d and smaller DTI values. Conclusions can be made that (1) a proper choice in d incorporating with the largest DTI (=6000 nm in this study) can elevate the *GR* values for these four wavelength regions; however the maximum GR for the wavelength region of 810~860 nm has its value much larger than those arising in the visible light wavelength regions for the RGB pixels; (2) for a fixed DTI value, an increase in *d* is favorable for the *GR* increase in the 810~860 nm wavelength region; however, is unfavorable for the *GR* increases in the regions of 450~460 nm (blue), 530~540 nm (green), and 610~620 nm (red) wavelengths; (3) the increasing rate of *GR* due to an increase in DTI becomes fairly small as the DTI depth is sufficiently large; this behavior is demonstrated in the present study as DTI = 5000~6000 nm is prepared.

## 4. Conclusions

In the present study, the photodiode in a RGB array was solved using the FDTD method in the Lumerical software to investigate its optical efficiency (OE) behavior affected by the combinations of various DTI depths and pitch sizes for the structure operating in a wide wavelength range of 300 to 1100 nm. Conclusions are made as follows:(1)With the geometries and materials reported in a literature and the optical properties of refraction index (*n*) and extinction coefficient (*k*) for a wide range of wavelengths, the simulational QE results predicted by the present model are very close to the experimental ones for the RGB pixels in the entire wavelength range. This model can be extended to evaluate the effects of various DTI and d combinations on the *OE_max._* values arising in the visible light (~300–700 nm) and NIR (~701–1100 nm) wavelength regions precisely.(2)Increasing the DTI can lead to monotonous *OE_max._* rises in the entire wavelength region, irrespective of the d value. A flat plane without the IPA structure (*d* = 0 nm) incorporating with DTI depth = 6000 nm is needed to produce the Max. *OE_max._* of every CF pixel in the visible light wavelength region. For a fixed DTI depth, the highest *OE_max_* value occurs at a fixed d value strongly dependent on the DTI depth and the wavelength region we specify. A flat plane is needed for the Max. *OE_max_* formed in the visible light wavelength regions, and a nonzero *d* is required for the Max. *OE_max_*_._ formed in the NIR wavelength region.(3)The GR of OE defined on the bases of the pixel arrays without and with the DTI + IPA structure is useful to evaluate the OE promotions in the three visible light and one NIR wavelength regions due to the uses of various DTI and d combinations. The max. GR for the peaks in the NIR range of 810–860 nm has its value (82.2%) much higher than those (4.9% for blue, 4.2% for green and 6.6% for red) arising in the RGB pixels operating in the visible light wavelength regions with their *OE_max._* The combinations of various nonzero DTI and d values can bring in a definitely positive GR value for the NIR wavelength region.

## Figures and Tables

**Figure 1 sensors-20-03062-f001:**
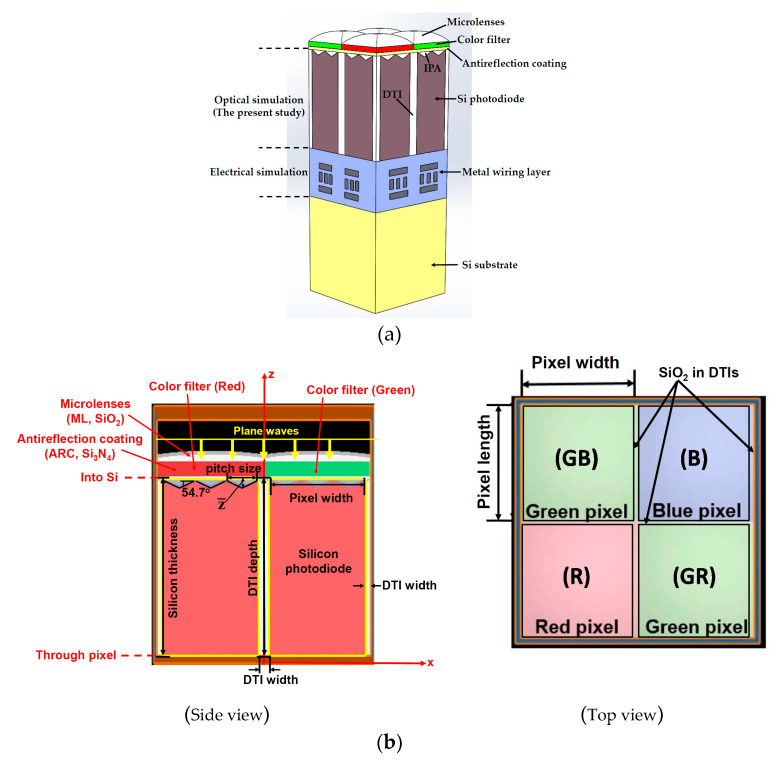
The schematic diagrams of (**a**) a backside-illuminated CMOS image sensor; and (**b**) the side and top view of a set of RGB pixels.

**Figure 2 sensors-20-03062-f002:**
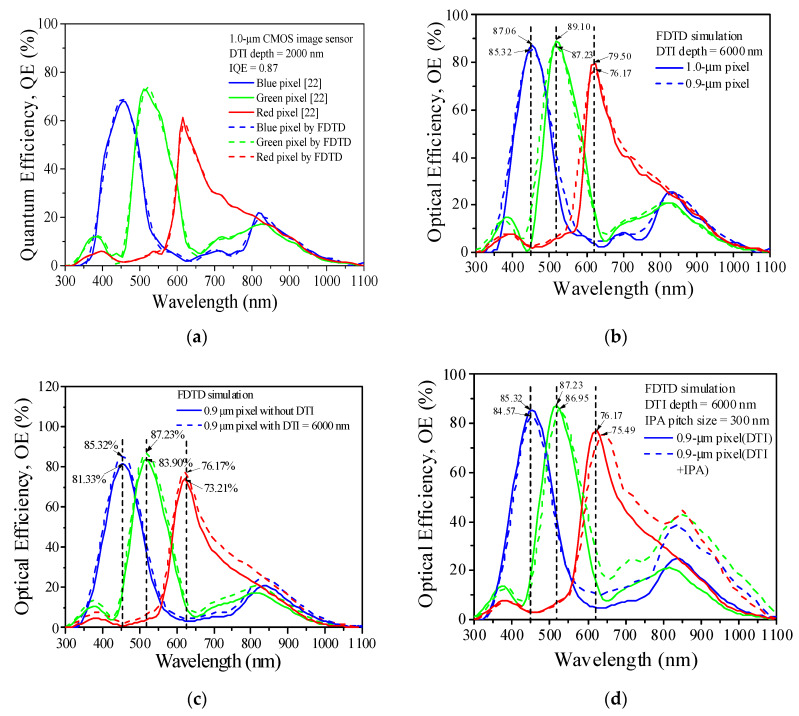
The evaluations of FDTD simulations for (**a**) the 1.0-μm pixel and QE results in [21]; and the OE results for (**b**) 1.0-μm and 0.9-μm pixels with DTI = 6000 nm; (**c**) 0.9-μm pixels without DTI and with DTI = 6000 nm; (**d**) 0.9-μm pixels with DTI and DTI + IPA structures.

**Figure 3 sensors-20-03062-f003:**
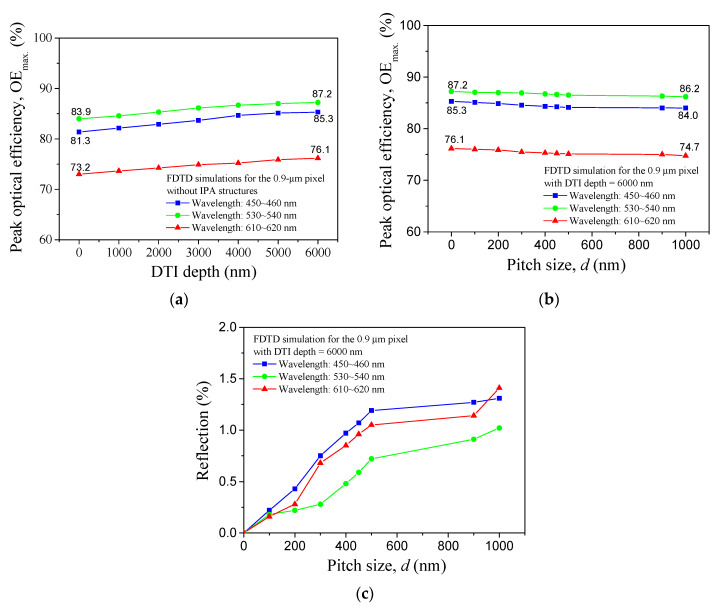
*OE_max_*_._ results created in the RGB pixels for various (**a**) DTI depth, and (**b**) pitch size values and in the visible light wavelength regions; (**c**) reflection (%) results created at various *d* values.

**Figure 4 sensors-20-03062-f004:**
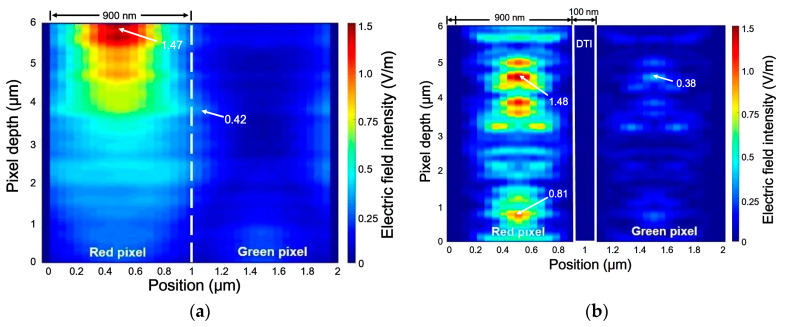
The cross-sectional distributions of electric field intensity for the 0.9-μm red and green pixels without the IPA structure but with (**a**) 0 nm, (**b**) 6000 nm as the DTI depth. They are obtained from using 850-nm incident waves.

**Figure 5 sensors-20-03062-f005:**
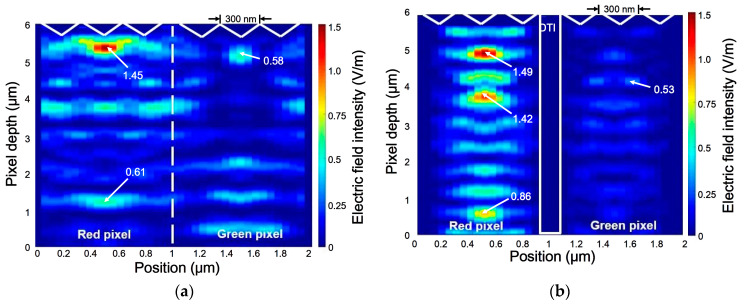
The cross-sectional distributions of electric field intensity for the 0.9-μm red and green pixels, with the IPA structure (*d* = 300 nm) and the DTI depth of (**a**) 0 nm; (**b**) 6000 nm. They are obtained from using 850-nm incident waves.

**Figure 6 sensors-20-03062-f006:**
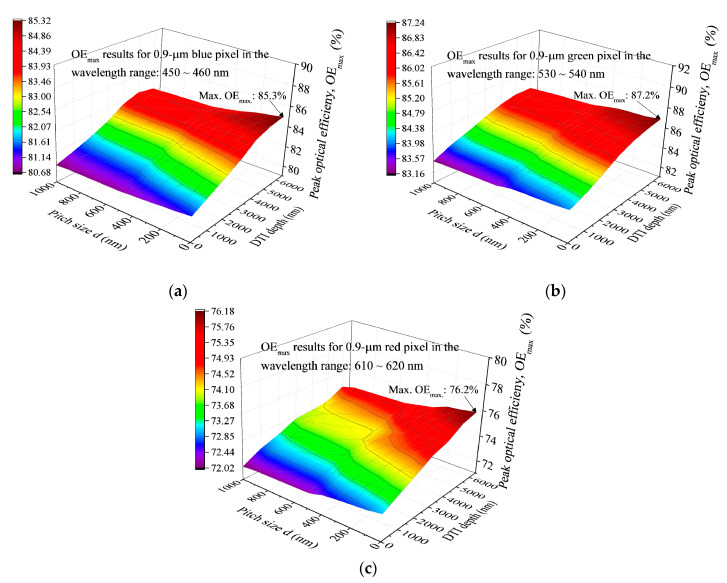
The *OE_max_* results predicted by the FDTD simulations for various DTI depths and pitch sizes in (**a**) blue pixel with 450~460 nm wavelengths; (**b**) green pixel with 530~540 nm wavelengths; (**c**) red pixel with 610~620 nm wavelengths; and (**d**) RGB pixels with 810~860 nm wavelengths.

**Figure 7 sensors-20-03062-f007:**
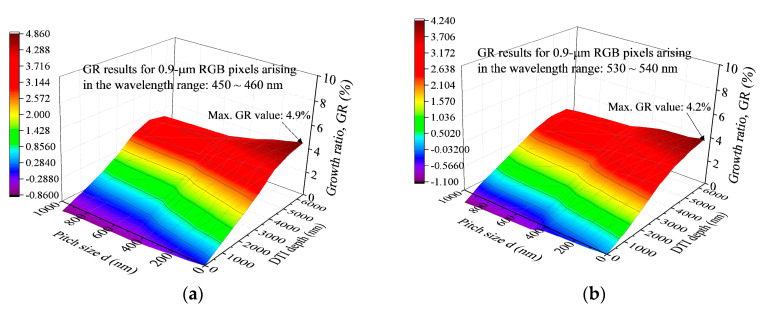
The GR results predicted by the FDTD simulations for various DTI depths and pitch sizes in (**a**) 450~460 nm (blue); (**b**) 530~540 nm (green); (**c**) 610~620 nm (red); and (**d**) 810~860 nm wavelength ranges.

**Table 1 sensors-20-03062-t001:** Optical parameters of materials in FDTD simulations for CMOS sensors [17,21,22].

Wavelength (nm)	Silicon Photodiode (Si) [17]		SiO_2_ Microlenses [17]		Si_3_N_4_ Antireflection Coating [17]		Blue Filter [21,22]		Green Filter [21,22]		Red Filter [21,22]	
	*n*	*k*	*n*	*k*	*n*	*k*	*n*	*k*	*n*	*k*	*n*	*k*
300	4.52	0.03	1.47	0	2.10	0	1.552	1.1 × 10^−5^	1.623	0.8× 10^-3^	1.538	8.3 × 10^-5^
400	5.57	0.39	1.47	0	2.08	0	1.549	1.4 × 10^−5^	1.601	1.0× 10^-3^	1.536	6.5 × 10^-5^
500	4.28	0.07	1.46	0	2.03	0	1.539	4.3 × 10^−5^	1.585	8.1× 10^−5^	1.532	3.1 × 10^−5^
600	3.90	0.03	1.45	0	2.01	0	1.534	1.6 × 10^−4^	1.576	1.0 × 10^−4^	1.529	9.7 × 10^−6^
700	3.78	0.01	1.45	0	2.00	0	1.531	9.0 × 10^−5^	1.572	2.2 × 10^−4^	1.528	6.6 × 10^−6^
800	3.69	0.07	1.45	0	1.99	0	1.529	9.1 × 10^−5^	1.569	2.1 × 10^−4^	1.527	7.5 × 10^−6^
900	3.61	4 × 10^−3^	1.45	0	1.99	0	1.528	9.3 × 10^−5^	1.567	2.2 × 10^−4^	1.526	3.7× 10^−6^
1000	3.58	5 × 10^−4^	1.45	0	1.98	0	1.527	7.9 × 10^−5^	1.565	1.8 × 10^−4^	1.526	2.3 × 10^−6^
1100	3.55	1 × 10^−4^	1.45	0	1.98	0	1.527	6.2 × 10^−5^	1.565	1.1 × 10^−4^	1.525	2.2 × 10^−6^

*n*: Index of refraction. *k*: Extinction coefficient

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
