# Peer review of "Deep Trench Isolation and Inverted Pyramid Array Structures Used to Enhance Optical Efficiency of Photodiode in CMOS Image Sensor via Simulations"

_sensors, 2020, doi:10.3390/s20113062_

Round 1

Reviewer 1 Report

The authors present, in this paper, simulations of CMOS based light sensors. The effect of various strategies such as Deep Trench Isolation and Inverted Pyramid Array Structures on the final efficiency of the sensors pixel have been assessed.

This paper is well structured and the model used is properly defined and validated with experimental data.

I suggest the following changes before publication:

  • The title is somehow misleading as it doesn’t reflect the fact that the present study is simulation based. As Sensors published both simulation and experimental study, the title should reflect this fact.
  • The schematics should be modifying to integrate more information. Information related to the geometry of the system are describe in the text and would be understand more easily with  visual help.
  • Please only add when you define term such as OPE, only make it once and not several times.
  • Line 129 add “in vacuum” to the definition of c
  • Fig 5 to 8: The font size should be increase on all the graphs

Author Response

Reply to Reviewers

Manuscript ID: sensors-805359

Title: Deep Trench Isolation and Inverted Pyramid Array Structures Used to Enhance Optical Efficiency of Photodiode in CMOS Image Sensor via Simulations

  This paper is well structured and the model used is properly defined and validated with experimental data.

I suggest the following changes before publication:

Reviewer’s Comment 1: The title is somehow misleading as it doesn’t reflect the fact that the present study is simulation based. As Sensors published both simulation and experimental study, the title should reflect this fact.

[Authors' Response] The Reviewer's comment is valuable and appreciated. We have corrected the title as "Deep Trench Isolation and Inverted Pyramid Array Structures Used to Enhance Optical Efficiency of Photodiode in CMOS Image Sensor via Simulations" in the revised manuscript.

Reviewer’s Comment 2: The schematics should be modifying to integrate more information. Information related to the geometry of the system are describe in the text and would be understand more easily with visual help.

[Authors' Response] The comment from Reviewer is appreciated. Figure 1(a) is newly added in the revised manuscript in order to describe the systematic diagram of the BSI CMOS image sensor. Figure 1(a) shows the schematic diagram of a backside-illuminated CMOS image sensor which is generally composed of the three parts: RGB pixels, metal wiring layer and Si substrate. They are illustrated in page 3, line 110 ~ line 111.

Reviewer’s Comment 3: Please only add when you define term such as OPE, only make it once and not several times.

[Authors' Response] The comment from Reviewer is appreciated. We have defined and checked the terms such as OE, DTI, and d only make each of them once in the revised manuscript.

Reviewer’s Comment 4: Line 129 add “in vacuum” to the definition of c

[Authors' Response] Section 2 has been rewritten such that the part Reviewer pointed out has been deleted from the revised version in accordance with according another Reviewer's comment. The comment from Reviewer is appreciated.

Reviewer’s Comment 5: Fig 5 to 8: The font size should be increase on all the graphs

[Authors' Response] The comment from Reviewer is appreciated. We have improved the font sizes of Fig.4 to Fig.7 in the revised manuscript to the best of own efforts. However, some font sizes of figures are unchanged due to the restriction of graph software.

All comments from Reviewers and Editor are deeply appreciated.

Reviewer 2 Report

This paper studies an image sensor including a deep trench isolation and an inverted pyramid array for an improve transmission in red. The paper is well written, but some results should be better analysed, and in my opinion it lacks a clear description of the state of the art, and what is the contribution of this work to the community. Then, the conclusion as to be improved and has to better highlight the novelty of this work.

Here are my remarks:

  • The concept of optical efficiency must be defined as it is not commonly used in the community. Do the authors mean transmission efficiency? Or Quantum Efficiency? If needed, please add a reference.
  • Page 1; line 35: what is the reason to speak about hybrid detectors? I don’t understand the link we this paper. Same comments of avalanche PDs and NIRS. These descriptions should be removed in my opinion.
  • Page 2, line 51: please justify that DTI is the best solution: data or reference
  • C-Si must be explained: is it crystalline silicon?
  • Page 2, line 56: the introduction of pyramid structures lacks an explanation of their utility. Why do we need then? For a better light focalisation? A better transmission factor?
  • Page 2, line 61: the sentence is not clear. Do the authors mean that pyramid structures have a longer interface than a flat surface? Therefore how can the pyramid structures minimize surface recombination effects if the interface is longer?
  • Page 2, line 81: authors say that optical performances have seldom reported. Can they give descriptions and references of these studies? Therefore, as similar studies seem to exist, what is the novelty of this study? Please comment and justify.
  • Page 2, line 85: why do authors need a 3D simulation? Would not be sufficient with a more simpler and efficient 2D simulation?
  • Page 3, section 2.1: this section is useless, as FDTD equations are known. Authors should put some references instead of this section.
  • Page 3, line 135: I don’t understand what are the two sides of the silicon substrate: the front and back side? What could be the aim of front side structures (SiO2, planar layer) if the device is back-illuminated?
  • Page 5, line 161: I don’t understand the “+” and square signs, as they are not shown in the Figure 1b. Do the authors mean that DTI are put all around pixels in a square shape? If yes, the explanation should be improved.
  • Page 5, line 168: why is the crystallographic direction (111) instead of (100)? Can you justify it?
  • Page 6, line 190: authors must explain what is the f-number
  • Page 6, line 197: the factors (DTI and d) are in contradiction with the “two factors” declared previously in Page 2, line 79. Please change the same or better clarify this nomenclature.
  • Page 6 line 208: do the authors neglect reflections by metal layers? If yes, I think their contribution is not negligible. Please, comment.
  • Page 6, line 213. I am not agree with the QE definition. QE is just a ratio of collected electrons over incident photons. Then, two different QE may be defined, the external one and the internal one. Then, image sensors deal with charge and potential after charge to voltage conversion in the sense node. Thus, the number of collected electrons is generally found from the sense node potential variation and the Charge to Voltage conversion Factor. As the authors extract the QE from a current, they must explain how is read the photodiode: is it a common readout circuitry (with source follower T, reset T, row select T) ? or is the photodiode photocurrent directly read?
  • Fig 3 c: I don’t understand how the optical efficiency (transmission??) can be improved by (lambda > 700nm) adding DTI structure. Is it due to a better control of the crosstalk and therefore to a better collection of red pixels? If yes, why is it no visible on other blue and green pixels?
  • Page 10 line 302: Lumerical instead of numerical?
  • Page 10, line 306: definition of EI?
  • Page 11: how can the authors explain the fact that OE increases with the d decrease? I cannot understand a pixel of d=0….how can you simulate this value?
  • Page 12, line 365: I don’t understand the meaning of the sentence. Is OE^DTI+IPA a new parameter? How is it evaluated? This paragraph should be improved for a better understanding.
  • Fig 7: what is the contribution of the Fig 7 compared to Fig8? It is not clear, and maybe the Fig 8 is enough according the conclusions. Please comment.
  • Conclusion: the sentence “d = 0 nm is needed to incorporate with DTI = 6000 nm to produce the Max” is not clear. Still, I don’t understand a pixel with a 0µm pitch. Authors do not conclude about the contribution of IPA: should it be used for future designs? Then, the main conclusion seems to be that DTI reduces the cross-talk, but this statement is already know and is not new. Please, improve the conclusion in order to highlight the novelty of this work.

Minor comments:

  • Table1: the resolution is two weak
  • Fig 2 could be removed as it is similar to Fig 1a. I recommend to replace Fig1a by Fig2.

Author Response

Reply to Reviewers

Manuscript ID: sensors-805359

Title: Deep Trench Isolation and Inverted Pyramid Array Structures Used to Enhance Optical Efficiency of Photodiode in CMOS Image Sensor via Simulations

Reviewer’s Comment 1: The concept of optical efficiency must be defined as it is not commonly used in the community. Do the authors mean transmission efficiency? Or Quantum Efficiency? If needed, please add a reference.

[Authors' Response] The comments and suggestion from Reviewer are appreciated. Figure 1(a) shows the schematic diagram of a backside-illuminated CMOS image sensor which is generally composed of the three parts: RGB pixels, metal wiring layer and Si substrate. The pixel transmission (T) part records the normal component of the Poynting vector, from the top surface (into Si photodiode) to the bottom surface (through the RGB pixels) of the Si photodiode (see Fig.1(b)). The Poynting vector ( unit: W/m-2) is obtained as the cross product of the electric and magnetic fields:

                                                               (1)

The simulator propagates the electric field in the primary grid and the magnetic field in the secondary grid. The steady simulational solution is used to interpolate the components of the magnetic field on the primary grid, making it possible to calculate the Poynting components everywhere inside the simulational region. If we want to quantify the overall optical performance, it is necessary to calculate the absorbed energy in the silicon photodiode. To calculate the power absorbed in each pixel, we integrate Poynting vector over the depletion region of the pixel. The optical efficiency (OE) is defined as the fraction of the power incident onto the pixel, that is absorbed in the depletion region of the pixel [17, 18]:

           (2)

The OE of a material can be obtained from the FDTD simulations [17]. (page 3, line 110 ~ line 126). Therefore, the OE is defined here including the absorption and transmission of incident waves in the Si photodiode, rather than the transmission only.

   QE is defined as the ratio of the collected electrons to the incoming photons, it is affected by the illumination conditions, the objective lens, the OE of the optical stack of the image sensor, and the collection efficiency of electronics. A full calculation of the QE involves both optical and electrical modelings. It is generally classified as the external quantum efficiency (EQE) and the internal quantum efficiency (IQE). The IQE is the ratio of collected charge number to the number of incident photons, and is dimensionless. In the electrical stimulation of Lumerical software [17], the IQE can be calculated by a Green's function approach for an arbitrary optical generation rate by determining a spatially-varying weighting function that represents the collection probability for a photo-generated electron-hole pair. Assuming that each absorbed photon generates an electron hole pair, EQE is simply the product solution of OE and IQE and it is written as [17]:

QE=EQE = IQEOE                                             (4)

The OE solutions can be obtained from the FDTD simulations if the controlling parameters including the dimensions of pixel array, DTI, d, and IQE are available. (page 6, line 216 ~ line 228). The comment from Reviewer is appreciated.

Reviewer’s Comment 2: Page 1; line 35: what is the reason to speak about hybrid detectors? I don’t understand the link we this paper. Same comments of avalanche PDs and NIRS. These descriptions should be removed in my opinion.

[Authors' Response] The comment and suggestion from Reviewer are appreciated. We have removed these reflections relevant to the avalanche PDs and NIRS in the revised manuscript.

Reviewer’s Comment 3: Page 2, line 51: please justify that DTI is the best solution: data or reference

[Authors' Response] The comment from Reviewer is appreciated. The revised version is changed as “The deep trench isolation (DTI) is a good approach for suppressing electrical crosstalk” (page 1, lin 43 to page 2, line 1).

Reviewer’s Comment 4: C-Si must be explained: is it crystalline silicon?

[Authors' Response] The "C-si" is defined as the crystalline silicon (page 2, line 55).

Reviewer’s Comment 5: Page 2, line 56: the introduction of pyramid structures lacks an explanation of their utility. Why do we need then? For a better light focalisation? A better transmission factor?

[Authors' Response] The comment from Reviewer is appreciated. Various light-trapping structures based on crystalline Si materials, including nanowires, nano rods and holes, and inverted pyramids, have been investigated through numerical simulation [8-10]. The nanowire arrays with moderate filling ratio have much lower reflectance compared to thin films [8] and nanohole arrays have great potential for efficient solar photovoltaics [9]. The effects of pyramid nanostructures correspond to a reduction in silicon mass by 2 orders of magnitude, pointing to the promising future of thin crystalline silicon solar cells [10]. Experimental results show that crystalline silicon (c-Si) thin films with a thickness of less than 10 μm can absorb light as well as 300-μm-thick flat c-Si substrates when they are decorated with inverted nanopyramid structures [11]. Other advantages of using inverted nanopyramids are that the interface area in the silicon photodiode is 1.7 times larger than that of a flat surface, which can minimize surface recombination losses, and that the fabrication of the structures is easy and cost-effective. They are illustrated in page 2, line 49 ~ line 60.

Reviewer’s Comment 6: Page 2, line 61: the sentence is not clear. Do the authors mean that pyramid structures have a longer interface than a flat surface? Therefore how can the pyramid structures minimize surface recombination effects if the interface is longer?

[Authors' Response] The advantages of using inverted nanopyramids are that the interface area in the silicon photodiode is 1.7 times larger than that of a flat surface, which can minimize surface recombination losses, and that the fabrication of the structures is easy and cost-effective (page 2, line 58 ~ line 60). The effect of pyramid structures on the optical efficiency (OE) parameters will be discussed more detailedly in the present study. The comment from Reviewer is appreciated.

Reviewer’s Comment 7: Page 2, line 81: authors say that optical performances have seldom reported. Can they give descriptions and references of these studies? Therefore, as similar studies seem to exist, what is the novelty of this study? Please comment and justify.

[Authors' Response] The DTI depth (DTI) and the pitch size (d) in the IPA structure can affect the crosstalk and photon collection inside the RGB pixels. The combined effect of these two controlling factors on optical efficiency (OE) in a wide range of wavelengths including the NIR region has seldom been reported [12]. The reported experimental and simulational results for visible light wavelengths are quite limited and insufficient to determine the optimal conditions of these two factors for the peak optical efficiency (OEmax). Numerical analyses made for the single-factor effect of DTI and IPA structures show that OEmax values in a wide wavelength ranges are elevated, and especially significant for those created in the NIR wavelengths. The finding of the optimum combination of DTI and IPA structures to obtain the maximum values of OEmax (Max OEmax) in these visible-light and NIR regions through the efficient and accurate way of numerical analyses becomes the purpose and novelty of this study. (page 2, line 76 ~ line 86).

  If the pitch size of the IPA structure on Si photodiode is much smaller than the wavelength of interest, the effective refractive index between Si photodiode and the upper layer has an intermediate value, which is well-known as an anti-reflection coating or refractive index gradient structure [20]. Meanwhile, a nonzero value is expected to have the diffraction and anti-reflection effects, thus elongating the optical path length within the Si layer [12]. An increment of d generally results in an absorption decrease and reflection increase slightly. (page 5, line 167 ~ line 173). The comment from Reviewer is appreciated.

Reviewer’s Comment 8: Page 2, line 85: why do authors need a 3D simulation? Would not be sufficient with a more simpler and efficient 2D simulation?

[Authors' Response] Reviewer’s comment is valuable. In this study, two green pixels in an array are positioned in a diagonal form to separate the red and blue pixels on their two sides. A three-dimensional (3D) solution for a Bayer filter array with the red/green/blue (RGB) pixels in the backside-illuminated CMOS image sensor was thus obtained using the FDTD Solutions software (page 2, line 86 ~ line 89).

Reviewer’s Comment 9: Page 3, section 2.1: this section is useless, as FDTD equations are known. Authors should put some references instead of this section.

[Authors' Response] The comment from Reviewer is appreciated. We have rewritten Sec. 2.1 and defined the optical efficiency (OE) to be more clear in the revised manuscript. The FDTD software [17] is usually applied to solve the time-dependent solutions for the Maxwell-Boltzmann differential equations. A change in the electric field (E) is linked with a change in the magnetic field (H) across the space. Generally, FDTD engines use the Yee algorithm to solve these equations [15, 19]. The periodic boundary condition considers the simulated structure to be infinitely repeated, like a periodic crystal, which allows us to reduce the simulated domain to its elementary zone. The present work adapts the FDTD Solutions software to develop a relevant pixel model while keeping the computational time and memory usage to be reasonable. The optical software Lumerical FDTD Solutions [17] was adopted because of its high efficiency as the calculation engine. Figure 1(a) shows the schematic diagram of a backside-illuminated CMOS image sensor which is generally composed of the three parts: RGB pixels, metal wiring layer and Si substrate. The pixel transmission (T) part records the normal component of the Poynting vector, from the top surface (into Si photodiode) to the bottom surface (through the RGB pixels) of the Si photodiode (see Fig.1(b)). The Poynting vector ( unit: W/m-2) is obtained as the cross product of the electric and magnetic fields:

                                                              (1)

The simulator propagates the electric field in the primary grid and the magnetic field in the secondary grid. The steady simulational solution is used to interpolate the components of the magnetic field on the primary grid, making it possible to calculate the Poynting components everywhere inside the simulational region. If we want to quantify the overall optical performance, it is necessary to calculate the absorbed energy in the silicon photodiode. To calculate the power absorbed in each pixel, we integrate Poynting vector over the depletion region of the pixel. The optical efficiency (OE) is defined as the fraction of the power incident onto the pixel, that is absorbed in the depletion region of the pixel [17, 18]:

           (2)

The OE of a material can be obtained from the FDTD simulations [17] (page 3, line 101 ~ line 126).

Reviewer’s Comment 10: Page 3, line 135: I don’t understand what are the two sides of the silicon substrate: the front and back side? What could be the aim of front side structures (SiO2, planar layer) if the device is backilluminated?

[Authors' Response] The comment from Reviewer is valued. Figure 1(a) is newly added in the revised manuscript in order to describe the systematic diagram of the BSI cmos image sensor. Figure 1(a) shows the schematic diagram of a backside-illuminated CMOS image sensor which is generally composed of the three parts: RGB pixels, metal wiring layer and Si substrate (page 3, line 110 ~ line 111). A RGB pixel array in a backside-illuminated CMOS image sensor (see Fig.1(a)) is adopted for the OE evaluations using the numerical scheme in Lumerical FDTD Solutions software for photon simulations. The simulations were carried out for the 3D domain of one RGB pixel array only. They are illustrated in page 4, line 152 ~ line 155.

Fig. 1(a)  The schematic diagrams of a backside-illuminated CMOS image sensor.

Reviewer’s Comment 11: Page 5, line 161: I don’t understand the “+” and square signs, as they are not shown in the Figure 1b. Do the authors mean that DTI are put all around pixels in a square shape? If yes, the explanation should be improved.

[Authors' Response] The DTI structures have been prepared in the areas with either a cross("+") form pattern to separate or a square form ("â–¡") to frame the four pixels, as shown in Fig. 1(b). The widths for the "+" and "â–¡" DTI structures were set to 100 and 50 nm, respectively.  The thicknesses for the components of microlenses, color filters, antireflection coating, and photodiode Si substrate are 360, 500, 50, and 6000 nm, respectively. (page 5, line 160 ~ line 164). The comment from Reviewer is appreciated.

Reviewer’s Comment 12: Page 5, line 168: why is the crystallographic direction (111) instead of (100)? Can you justify it?

[Authors' Response] This mistake has been corrected. In the simulations, Si (100) is adopted as the crystallographic direction of the surface before etching (page 5, line 173 ~ line 174).

Reviewer’s Comment 13: Page 6, line 190: authors must explain what is the f-number

[Authors' Response] Light source is generally produced by a halogen lamp and a monochromator. The light source was partly guided to a calibrated photodiode to control its intensity. A f-number diffuse light incident on the sensor can be created via the light passing through a diffuser and a f/D controller. In this study, the diffuser is assumed to be a sufficiently large size such that every pixel in the sensor is expressed to a uniform illumination with the same f value in the simulations. Then, only the "Lumerical" module in the FDTD software meets this requirement (page 6, line 192 ~ line 197). The comment from Reviewer is appreciated.

Reviewer’s Comment 14: Page 6, line 197: the factors (DTI and d) are in contradiction with the “two factors” declared previously in Page 2, line 79. Please change the same or better clarify this nomenclature.

[Authors' Response] The simulation scheme was applied to obtain the OE solutions and investigate the effects of the controlling factors (DTI and d) on OE. (page 6, line 201 ~ line 202). We have improved and used the same nomenclatures in the revised manuscript. The comment from Reviewer is appreciated.

Reviewer’s Comment 15: Page 6 line 208: do the authors neglect reflections by metal layers? If yes, I think their contribution is not negligible. Please, comment.

[Authors' Response] In the present study, we focus on the effects of combing different DTI and d of the IPA structure on the OEs arising in the silicon photodiode layer of the RGB pixels only. The effect of the reflection of the transmission by the metal wiring layer below the photodiode will be discussed in the later section. Therefore, the FDTD simulations are carried out for the photodiode domains (excluding the microlenses and antireflection coating) whose top surface is exposed to the plane waves (page 3, line 137 to page 4, line 142).

  In the FDTD analyses, a uniaxial anisotropic perfectly matched layer was assumed for the boundaries normal to the z direction (see Fig.1(b)). This assumption was made to avoid the multiple reflections from the pixel components above the Si photodiode. Regarding the light passing through the Si photodiode, the normalized transmission () in the position of “through the pixels” is evaluated to have the quantify about 0.5 ~ 0.8% of the quantify of “into the Si photodiode” in the FDTD simulational results. Thus, the optical reflection by the metal wiring layer below the Si photodiode can be neglected. (page 6, line 205 ~ line 211). The comment from Reviewer is valuable.

Reviewer’s Comment 16: Page 6, line 213. I am not agree with the QE definition. QE is just a ratio of collected electrons over incident photons. Then, two different QE may be defined, the external one and the internal one. Then, image sensors deal with charge and potential after charge to voltage conversion in the sense node. Thus, the number of collected electrons is generally found from the sense node potential variation and the Charge to Voltage conversion Factor. As the authors extract the QE from a current, they must explain how is read the photodiode: is it a common readout circuitry (with source follower T, reset T, row select T) ? or is the photodiode photocurrent directly read?

[Authors' Response] Reviewer’s comment is valuable. The explanation for the quantum efficiency parameter has been provided as “QE is defined as the ratio of the collected electrons to the incoming photons, it is affected by the illumination conditions, the objective lens, the OE of the optical stack of the image sensor, and the collection efficiency of electronics. A full calculation of the QE involves both optical and electrical modelings. It is generally classified as the external quantum efficiency (EQE) and the internal quantum efficiency (IQE). The IQE is the ratio of collected charge number to the number of incident photons, and is dimensionless. In the electrical stimulation of Lumerical software [17], the IQE can be calculated by a Green's function approach for an arbitrary optical generation rate by determining a spatially-varying weighting function that represents the collection probability for a photo-generated electron-hole pair. Assuming that each absorbed photon generates an electron hole pair, EQE is simply the product solution of OE and IQE and it is written as [17]:

    QE=EQE = IQEOE                                             (4)

The OE solutions can be obtained from the FDTD simulations if the controlling parameters including the dimensions of pixel array, DTI, d, and IQE are available (page 6, line 216 ~ line 228).

Reviewer’s Comment 17: Fig 3 c: I don’t understand how the optical efficiency (transmission??) can be improved by (lambda > 700nm) adding DTI structure. Is it due to a better control of the crosstalk and therefore to a better collection of red pixels? If yes, why is it no visible on other blue and green pixels?

[Authors' Response] Figure 2(c) shows the OE profiles for the RGB pixels in the 0.9-μm-pixel arrays with the solid curves defined for the pixel array without DTI and the dashed curves for the pixel array with DTI = 6000 nm. The nonzero DTI depth can elevate the OEmax. values for the blue, green and red pixels arising in the visible light wavelength region (OEmax.: 81.33 → 85.32% for blue, 82.90 → 87.23% for green, and 73.21 → 76.17% for red), and the OEmax. values of these three CF pixels operating in the NIR wavelength region (> 780 nm) (page 7, line 252 to page 8, line 257). Obviously, the increasing ratios for these three CF pixels are quite close. The comment from Reviewer is appreciated.

Reviewer’s Comment 18: Page 10 line 302: Lumerical instead of numerical?

[Authors' Response] We have corrected as "Lumerical" (page 10, line 306).

Reviewer’s Comment 19: Page 10, line 306: definition of EI?

[Authors' Response] The EI is defined as electric field intensity (page 9, line 303)

Reviewer’s Comment 20: Page 11: how can the authors explain the fact that OE increases with the d decrease? I cannot understand a pixel of d=0….how can you simulate this value?

[Authors' Response] d = 0 nm is defined for the top surface of silicon photodiode to be a flat plane (without IPA structure) perpendicular to the plane waves. In order to explain the OEmax. behavior demonstrated in Fig.3(b), the reflection created in the Si photodiode layer can be evaluated as a function of d. Figure 3(c) shows the reflection results in these three wavelength regions. Reflections are monotonically elevated by increasing the pitch size although their value is quite small (< 1.5%). An increment of reflection (%) will bring in the transmission and absorption decreases, thus resulting in the OEmax. decline (page 9, line 283 ~ line 289). The comment from Reviewer is appreciated.

Fig.3(c) Reflection (%) results created at various d values.

Reviewer’s Comment 21: Page 12, line 365: I don’t understand the meaning of the sentence. Is OE^DTI+IPA a new parameter? How is it evaluated? This paragraph should be improved for a better understanding.

[Authors' Response] The OEmax. data shown in Fig.6 are provided to evaluate the growth ratio (GR) of optical efficiency (OE) due to the use of different d and DTI. Define SOEDTI+IPA as the sum of the OE values arising in a specified wavelength range for the pixel array with various combinations of nonzero DTI and d. For example, the SOEDTI+IPA value in the 450 ~ 460 nm wavelength region is obtained to be the sum of ()blue and the OEDTI+IPA values of the green and red pixels evaluated at the same wavelength. SOENo DTI+IPA is also defined as the sum of the OE values similar to these defined for SOEDTI+IPA unless it is obtained for the array without the DTI and IPA structures. These two SOE parameters are defined valid for the wavelengths in the 450-460 nm, 530-540 nm, and 610-620 nm regions only. As to the SOEDTI+IPA and SOENo DTI+IPA parameters defined for the wavelength range between 810 and 860 nm, they are obtained to be the sum of the three OEmax. values of the RGB pixels existing in this wavelength ­region. (page 12, line 365 ~ line 375). The comment from Reviewer is appreciated.

Reviewer’s Comment 22: Fig 7: what is the contribution of the Fig 7 compared to Fig8? It is not clear, and maybe the Fig 8 is enough according the conclusions. Please comment.

[Authors' Response] The GR value obtained from these two SOE values shows its definition basis characteristically different from those the OEmax results shown in Fig.6. (page 12, line 377 ~ line 378). Therefore, Figure 7 in the revised manuscript is still retained. The comment from Reviewer is appreciated.

Reviewer’s Comment 23: Conclusion: the sentence “d = 0 nm is needed to incorporate with DTI = 6000 nm to produce the Max” is not clear. Still, I don’t understand a pixel with a 0μm pitch. Authors do not conclude about the contribution of IPA: should it be used for future designs? Then, the main conclusion seems to be that DTI reduces the cross-talk, but this statement is already know and is not new. Please, improve the conclusion in order to highlight the novelty of this work.

[Authors' Response] Increasing the DTI can lead to monotonous OEmax. rises in the entire wavelength region, irrespective of the d value. A flat plane without the IPA structure (d = 0 nm) incorporating with DTI depth = 6000 nm is needed to produce the Max. OEmax. of every CF pixel in the visible light wavelength region. For a fixed DTI depth, the highest OEmax­ value occurs at a fixed d value strongly dependent on the DTI depth and the wavelength region we specify. A flat plane is needed for the Max. OEmax formed in the visible light wavelength regions, and a nonzero d is required for the Max. OEmax. formed in the NIR wavelength region (page 13, line 413 to page 14, line 419). The comment from Reviewer is appreciated.

Reviewer’s Comment 24: Minor comments:

Table1: the resolution is two weak

[Authors' Response] We have improved the resolutions of Table 1 (page 5, line 189). The comment from Reviewer is appreciated.

Reviewer’s Comment 25: Fig 2 could be removed as it is similar to Fig 1a. I recommend to replace Fig1a by Fig2.

[Authors' Response] The comment and suggestion from Reviewer are appreciated. The Fig1(b) in the revised version has been replaced by Fig.2 in the old manuscript.

All comments from Reviewers and Editor are deeply appreciated.

Round 2

Reviewer 2 Report

The manuscript has been improved. I just have a minor comment:

  • Equation (2) : source instead of soruce

Author Response

Reply to Reviewers

Manuscript ID: sensors-805359

Title: Deep Trench Isolation and Inverted Pyramid Array Structures Used to Enhance Optical Efficiency of Photodiode in CMOS Image Sensor via Simulations

  The manuscript has been improved. I just have a minor comment:

Reviewer’s Comment 1: Equation (2) : source instead of soruce

[Authors' Response] We have corrected this spelling errors and underlined in the revised manuscript (page 3, line 125). The comment from Reviewer is appreciated.

All comments from Reviewers and Editor are deeply appreciated.
